# Analysis of Co-Crystallization Mechanism of Theophylline and Citric Acid from Raman Investigations in Pseudo Polymorphic Forms Obtained by Different Synthesis Methods

**DOI:** 10.3390/molecules28041605

**Published:** 2023-02-07

**Authors:** Yannick Guinet, Laurent Paccou, Alain Hédoux

**Affiliations:** UMR 8207, UMET—Unité Matériaux et Transformations, University Lille, CNRS, INRAE, Centrale Lille, F-59000 Lille, France

**Keywords:** Raman spectroscopy, physical stability, co-milling, H-bonding, π-H-bonding

## Abstract

Designing co-crystals can be considered as a commonly used strategy to improve the bioavailability of many low molecular weight drug candidates. The present study has revealed the existence of three pseudo polymorphic forms of theophylline–citric acid (TP–CA) co-crystal obtained via different routes of synthesis. These forms are characterized by different degrees of stability in relation with the strength of intermolecular forces responsible for the co-crystalline cohesion. Combining low- and high-frequency Raman investigations made it possible to identify anhydrous and hydrate forms of theophylline–citric acid co-crystals depending on the preparation method. It was shown that the easiest form to synthesize (form 1′), by milling one hydrate with an anhydrous reactant, is very metastable, and transforms into the anhydrous form 1 upon heating or into the hydrated form 2 when it is exposed to humidity. Raman investigations performed in situ during the co-crystallization of forms 1 and 2 have shown that two different types of H-bonding ensure the co-crystalline cohesion depending on the presence of water. In the hydrated form 2, the cohesive forces are related to strong O–H … O H-bonds between water molecules and the reactants. In the anhydrous form 1, the co-crystalline cohesion is ensured by very weak H-bonds between the two anhydrous reactants, interpreted as corresponding to π-H-bonding. The very weak strength of the cohesive forces in form 1 explains the difficulty to directly synthesize the anhydrous co-crystal.

## 1. Introduction

It is well recognized that most of the pharmaceutical products composed of small active molecules are stored and ingested in the solid-state [1]. Additionally, they are mainly synthesized in the crystalline state often characterized by poor water solubility associated with low bioavailability [2] and consequently are not very efficient. Different strategies have been developed to overcome the poor water solubility of active pharmaceutical ingredients (API) [3]. Designing co-crystals can be considered a recently growing technique, from the last two decades, to enhance dissolution rates and bioavailability [4,5]. It is based on the fundamentals of crystal engineering [6] via supramolecular synthons [7]. In this context, designing co-crystals is based on the formation of supramolecular hydrogen bonded networks or π-hydrogen bonded complexes. The π–π interactions also have an influence on structural organization [8], generally to a lesser extent than H-bonding. Carboxylic acid is probably the most commonly used functional moiety in crystal engineering [5,6] to form supramolecular synthons, and it is widely involved in the formation of co-crystals. Consequently, citric acid composed of three carboxylic groups can be considered a basic co-former for obtaining a co-crystal. Indeed, it has been used to form co-crystals with theophylline and caffeine [9].

Polymorphism is crucial in pharmaceutical applications [10,11] because different crystalline forms have different physical properties, including solubility and bioavailability. Consequently, polymorphism has systematically focused investigations in single-component molecules in order to select the form in which the active pharmaceutical ingredient (API) is the most effective. In multi-component systems, e.g., in co-crystals, polymorphic situations are relatively unexplored, while they are used to improve physical properties, especially the solubility of the APIs. The presence of multiple functional groups inherent to the formation co-crystals can generate supramolecular synthon polymorphism [6] responsible for polymorphism in co-crystals [12].

Co-crystals can be synthesized by a variety of methods [5], i.e., slow evaporation from solution, sublimation, growth from the melt, solid-state milling, hot melt extrusion and spray crystallization, which can be roughly classified as solid-state or solution based methods. Interestingly, it was found that the technique of solvent-drop grinding significantly improves the kinetics of co-crystallization [13]. It has most often admitted that the phase obtained is independent of the method used for the preparation of co-crystals [5]. However, it has been reported that the technique of solvent-drop grinding gives the opportunity to control the co-crystal polymorph in the caffeine–glutaric acid, by selecting the type of solvent (polar or non-polar) [14]. It was also found that the synthesis of caffeine–citric acid by evaporation from a solution [8] leads to a co-crystalline form different from that obtained by the drop-solvent grinding [9]. It has been recognized that various synthesis methods lead to various forms of co-crystals [8,12,15,16], associated with different molecular associations [5,6]. Intriguingly, the inability to reproduce a polymorphic form for caffeine–citric acid co-crystal was recently reported [17]. However, the relative stability of polymorphic forms and their phase transformations are rarely investigated [15].

The aim of the present study focused on the analysis of the crystalline forms (structure, H-bonding and stability) of co-crystals obtained by different routes in order to determine the mechanism of co-crystallization. Theophylline (TP, C_7_H_8_N_4_O_2_, Figure 1a) was selected as the API and citric acid (CA, C_6_H_8_O_7_, Figure 1b) as the co-former. It has been recognized that TP undergoes reversible anhydrous–hydrate transformations with direct consequence on its solubility and its bioavailability [18,19], and it was shown that co-crystallization was a way to enhance the physical stability of TP [20]. It is also well known that TP and CA easily form co-crystals anhydrate or hydrate, depending on the preparation and using individual components anhydrate or hydrate [9]. Figure 1 clearly shows the possibility to form H-bonds between TP and CA favoring the formation of complex structures. Special attention has been paid to the influence on H-bonding in the co-crystal formation and stability. The formation of co-crystal hydrates will be carefully analyzed, as well as the relative stability of anhydrous and hydrate co-crystals.

Low-frequency Raman spectroscopy investigations on ibuprofen (S/RS)—nicotinamide co-crystals [15] have recently revealed different forms of co-crystals depending on the preparation method, and very subtle polymorphic changes detected by varying the temperature. Given the sensitivity of Raman spectroscopy to detect polymorphic modifications and for detecting H-bonding [15], Raman spectroscopy has been widely used in this study to simultaneously analyze the crystalline form, the molecular conformation and H-bonding in TP–CA co-crystals. The important contribution of data provided by combining low- and high-frequency spectroscopy measurements for describing molecular interactions responsible for the stability of co-crystals has also been shown previously [21]. In the present study, the Raman spectra of the different forms of TP–CA co-crystals obtained by the net co-milling of anhydrous/hydrate reactants and the liquid-assisted milling of anhydrous components were presented. Their stability and transformation upon heating and RH exposition were then analyzed in order to better describe the molecular interactions responsible for the co-crystalline cohesion, via the determination of H-bond breaking. In this context, neat milling procedures of anhydrous reactants were also performed at room temperature and low temperature (cryogenic milling) for providing as much information as possible about the influence of milling on the H-bond formation between TP and CA.

## 2. Results

### 2.1. Identification of Co-Crystals after Preparation

In the first stage, the low- and high-frequency spectra of various types of TP–CA co-crystals synthesized via different methods were compared with those of anhydrous (TPa, CAa) and hydrated (TPh, CAh) individual components in Figure 2a,b. Two crystalline forms were obtained by co-milling. The co-crystal that presents an anhydrous character was then called form 1′ (as a distinction to the rigorous anhydrous form noted form 1); it was obtained by neat co-milling (i) TPa and CAh, (ii) TPh and CAa, (iii) or by liquid-assisted co-milling TPa and CAa. Co-crystal hydrate, called form 2, was obtained either by hydration of form 1′ (form 1′—hydration → form 2), or after being exposed to 98% relative humidity (RH) by keeping form 1′ in a desiccator containing a saturated solution of potassium sulfate for 72 h or within a RH cell for Raman investigations.

A low-frequency Raman spectrum (LFRS) is composed of the phonon peaks, i.e., the lattice modes, usually lying between 5 to 150 cm^−1^ for molecular materials. This spectrum represents the crystalline fingerprint of a crystalline form, making this technique an indirect structural probe. It is worth noting that phonon peaks in the spectrum of the hydrate co-crystal are clearly sharper than those of the anhydrous co-crystal. Apart from this feature, there is no strong difference between the LFRS of the co-crystalline forms. Common features can even be found, especially the two shoulders of the most intense band at 28 and 43 cm^−1^ in the spectrum of form 1′ corresponding to the well-defined bands in the spectrum of the hydrate form 2. There is also a correspondence of these shoulders with two phonon peaks in the spectrum of CA hydrate, indicating that these shoulders are the signature of the presence of a small amount of water. Consequently, the relationship between the two LFRS of co-crystals observed in Figure 2a reveals that the crystalline lattice of the hydrate co-crystal is not strongly different from that of the anhydrous form and the presence of water significantly favors ordering and probably co-crystallization.

The spectral region lying between 2800 and 3600 cm^−1^, plotted in Figure 2b, is the domain of X–H (X = C, O) stretching vibrations. It is worth noting that the high-frequency spectra of the commercial form (form II) of TPa and CAa are found to be in agreement with those reported in the literature [22,23]. As observed in the low-frequency region, C–H stretching bands lying between 2800 and 3000 cm^−1^ are also broader in the anhydrous co-crystal spectrum, as observed for the lattice modes. This indicates a lower degree of molecular organization in the anhydrous form. The 3050–3600 cm^−1^ range, mostly corresponding to the O–H stretching region, is plotted in Figure 2c for greater clarity in the assignment of the vibrational bands. The O–H stretching spectra of TP and CA hydrates are characterized by very broad and weakly intense bands lying between 3240 and 3460 cm^−1^, also observed in the spectrum of the co-crystal form 2. By contrast, these two bands do not exist in the spectrum of the anhydrous form. The spectrum of the co-crystal form 1′ can be easily distinguished by the presence of the relatively sharp band at 3510 cm^−1^ assigned to O–H stretching vibration in CA, from a quick look at Figure 2b,c. The presence of this band only in the co-crystal anhydrate suggests that this band could be associated with an intermolecular interaction that ensures the co-crystalline cohesion. The high-frequency of this band reflects very weak molecular interactions that could be the signature of inter molecular π-H-bonds between CA and TP. Indeed, π-electrons in aromatic rings can play the role of proton acceptor, forming very weak interactions [24,25,26]. This type of hydrogen bond is much weaker than typical σ-hydrogen bonds and can be easily perturbed by the environmental conditions, e.g., temperature and humidity. This can explain the absence of the 3510 cm^−1^ band in form 2 (see Figure 2c). This interpretation can be supported by the observation of this band at a higher frequency than the band reflecting H-bonding in CA anhydrate and hydrate, reflecting weaker molecular interactions between CA and TP than between CA molecules. The cohesion of the co-crystal form 2 is due to stronger O–H … O H-bonds via water molecules interacting between TP and CA, corresponding to the broad bands located at lower frequencies between 3240 and 3460 cm^−1^. The spectral region around 3100 cm^−1^ was assigned to C–H stretching vibrational bands [27]. This region makes it possible to distinguish the two co-crystal forms. The sharp band located around 3120 cm^−1^ in TPa exhibits a significant shift down to about 3110 cm^−1^ in TPh, interpreted as resulting from the formation of the C–H … O H-bond between theophylline and water [27]. Consequently, this spectral range is well suited for monitoring the hydration of theophylline as well as the hydration of TP–CA co-crystals. The opposite shift is observed between form 1′ and form 2. This analysis shows that the spectra of forms 1′ and 2 are more different in the high frequency range than in the low-frequency region. This clearly reveals the existence of two different types of H-bonding in the two co-crystal forms in relation to two different co-crystallization mechanisms.

### 2.2. Thermal Stability of Co-Crystals

The thermal stability of co-crystal forms 1′ and 2 was analyzed from the Raman spectra collected in the low- and high-frequency (respectively, noted LF and HF) regions along heating ramps ($\dot{T}=1\;deg/min$).

The temperature dependences of the LFRS and the O–H stretching region taken in the co-crystal form 1′ were plotted in Figure 3a,b, respectively. Figure 3a,b show the absence of significant modification of spectra in both regions upon heating. However, a closer look at Figure 3a allows detecting the disappearance of the shoulder at 28 cm^−1^ above 50 °C (located by the blue arrow), indicating the escape of the small amount of water. This indicates the relative metastability of the co-crystal form 1′ and indicates that it is not rigorously anhydrous.

The same investigations were performed on the co-crystal form 2, and the spectra taken in the LF and HF regions upon heating were plotted in Figure 4a,b. Distinction to form 1′, significant changes were observed both in the LF and HF regions.

In the LF region, changes can be mostly identified by the frequency shift of some lattice modes around 75 °C, accompanied by an increase in the intensity in the very LF domain (located by the thick blue arrow in Figure 4a, ω < 15 cm^−1^) usually considered as quasielastic intensity inherent to non-collective fast motions. This quasielastic intensity is detected only in a limited temperature range extending between 75 and 120 °C and can be interpreted as disordering induced by the water escape, and is not observed upon heating the co-crystal form 1′ (Figure 3a). At high temperatures (T > 100 °C), LF and HF spectra taken upon heating the hydrate form 2 resemble those of form 1, and collected upon heating the co-crystal form 1′, as shown in Figure 5. Figure 5a indicates a frequency shift between forms 1′ and 2 heated at 120 °C, with only that below 120 cm^−1^ corresponding to the domain of lattice modes. These shifts of phonon peaks could be induced by very weak differences between the lattice parameters correlated by different degrees of ordering in the molecular organization after water escape. A quasielastic intensity can also be detected in the LF spectrum of the heated form 2 that does not exist in the spectrum of form 1′ heated at 120 °C. These LF features can be considered as signatures of disorder induced by the water escape. This disordering can be confirmed by Figure 5b which shows a broadening of the 3510 cm^−1^ band in the spectrum of the heated form 2 as the only difference between the two spectra.

A closer inspection of Figure 4 reveals that the spectral changes in the LF and HF spectral regions are shifted from about 75 °C in the LF region to 100 °C in the HF region. This indicates that the lattice modifications are necessary for making possible the water escape. The analysis was then focused on Raman features identified by arrows in Figure 3 and Figure 4 to obtain more detailed information. The temperature dependences of the quasielastic intensity (I_QES_) and the frequency of phonon peaks located at 35 and 65 cm^−1^ were plotted for forms 1′ and 2 in Figure 6. As expected from a quick look at Figure 3a and Figure 4a, the most significant changes are observed upon heating form 2 around 75 °C. Surprisingly, very weak changes were also detected in form 1′ around 40 °C, i.e., at a lower temperature than those observed in the co-crystal hydrate. If the interpretation of the changes in the LFRS as being inherent to the water escape is adopted, Figure 6 confirms that co-crystal form 1′ contains a very small number of water molecules not or weakly bound to TP or/and CA molecules.

Deeper investigations in the HF region were carried out via the analyses of the frequency of the 3125 cm^−1^ Raman band and the intensity of Raman bands at 3360 and 3510 cm^−1^. Their temperature dependences were plotted in Figure 7. The 3135 cm^−1^ band in form 2 drops down to a position very close to that of form 1′ at 75 °C. Figure 7b clearly shows the intensity decrease in the 3360 cm^−1^ band, previously assigned to O–H stretching of water molecules associated with CA molecules, which allows the direct monitoring of the water escape (intensity at 3360 cm^−1^) and the development of new intermolecular forces (intensity at 3510 cm^−1^). Simultaneously to this intensity decrease, the intensity increase in the 3510 cm^−1^ band is clearly observed indicating the transformation of form 2 into the anhydrous form. Consequently, it was shown that forms 1′ and 2 transform into an anhydrous form 1 upon heating. It is clear that the structural organizations are very similar in form 1′ and 1, since low-frequency spectral modifications are almost undetectable without data refinement. Additionally, the co-crystalline cohesion is also similar and due to weak H-bonding detected by the high-frequency O–H stretching band located at 3510 cm^−1^. The positive temperature dependence of this band position (Appendix A) confirms the existence of molecular associations between TP and CA molecules. By contrast to co-crystal form 1′, the transformation of form hydrate 2 into the anhydrous form 1 is accompanied with significant high-frequency modifications reflecting changes in molecular associations detected via the disappearance of the two broad O–H stretching bands located between 3240 and 3260 cm^−1^, at the expense of the emerging O–H stretching band located at 3510 cm^−1^ (see Figure 7b). It is noticeable that only slight low-frequency modifications were observed accompanying the transformation of form 2 into form 1, reflecting only a light lattice change.

Figure 7b, highlights first the subtle but significant changes in the 3360 cm^−1^ band intensity at about 75 °C, corresponding to the detection of lattice changes. The two-step intensity decrease in this band can be related to the two-step intensity increase in the quasielastic intensity (Figure 6a) interpreted as resulting from the disordering induced by the water escape. This allows for the refinement of the description of the dehydration mechanism of form 2 as a two-step process: (i) the removal of a small number of water molecules requiring weak lattice changes and (ii) the faster escape of all the remaining water molecules.

### 2.3. Co-Milling Anydrous Reactants

#### 2.3.1. Cryogenic Co-Milling

Additional experiments were performed to obtain co-crystals by co-milling anhydrous forms of TP and CA at room temperature without success by co-milling at room temperature for 30 min. However, by cryogenic co-milling TPa and CAa a single phase co-amorphous mixture with equimolar ratio was obtained. The low-frequency spectra of this co-amorphous formulation were collected upon heating from −50 °C up to 180 °C at 1 °C/min. The temperature dependence of the low-frequency spectrum and I_QES_(T) were plotted in Figure 8.

Figure 8a clearly shows the amorphous band-shape of the low-frequency spectrum at −50 °C obtained by cooling the formulation just after cryo-milling. The absence of phonon peaks indicates that the mixture of anhydrous forms of TP and CA was amorphized by co-milling at low-temperature. Significant changes in the low-frequency spectrum reflect the phase transition sequence of the co-amorphous formulation upon heating, which can be better visualized by plotting the temperature dependence of the quasielastic intensity (I_QES_) in Figure 8b. The first feature corresponds to a change in the slope of I_QES_(T) usually assigned to the glass transition [28,29], observed at T_g_ = 10 °C. The second corresponds to a drop in intensity induced by crystallization (ordering process) at T_c_ = 70 °C into the anhydrous form 1. The spectra taken above 70 °C can be identified as the spectrum the anhydrous co-crystal form 1obtained by heating form 1′. The last feature is the increase in I_QES_ associated with the melting of the co-crystal around T_m_ = 185 °C.

#### 2.3.2. Co-Milling at Room Temperature for 60 min

It was reported that co-milling anhydrous TPa and CAa for 1 h makes it possible to obtain a co-crystal [9]. The LFRS of the sample obtained by the neat co-milling of anhydrous reactants is plotted in Figure 9a and compared with the spectrum of the anhydrous co-crystal form 1 obtained by recrystallization of the TP-CA amorphous blend, as above. The spectra are significantly different, with a strong low-frequency intensity enhancement in the mixture milled for 1 h, distinctive of a substantial disorder mimicking the spectrum of an amorphous molecular material. Additionally, the LFRS is composed of weakly intense and broad phonon peaks indicating an incomplete amorphization process after 1 h of milling. Consequently, increasing the milling time of TPa and CAa promotes disorder until co-amorphization, as observed by cryogenic milling. The broadened phonon peaks, corresponding to the not completely amorphized materials, seem to result from the two individual spectra of TPa and CAa. Surprisingly, it can be observed that the amorphous contribution spontaneously decreases in favor of a crystalline state progressively transforming into the anhydrous co-crystal form 1. Consequently, the neat co-milling of anhydrous TP and CA at room temperature does not directly provide co-crystal. The co-crystal formation from anhydrous reactants requires following a route necessarily passing through a transient disordered state. It can be observed that the transformation of the LFRS reflects a rapid ordering process (a decreasing of the amorphous contribution) in the early stages of the transformation, contrasting the formation of the anhydrous co-crystal achieved after about 96 h.

Raman investigations were simultaneously carried out in the high frequency region to better understand the mechanism of co-crystal formation. Spectra were plotted in Figure 9b at various times until 20 h of transformation, the complete monitoring of the transformation not being possible because of the too long transformation time. Spectra collected in the mixture milled for 1 h were compared with those of TPa and CAa. The first spectral modification in the high-frequency range is observed 3 h after the milling procedure corresponding to the emergence of the 3510 cm^−1^ band at the expense of the O–H stretching band in CAa. This observation shows the very slow formation of very weak intermolecular forces, interpreted as π-H-bond interactions. It is worth noting that the formation of these molecular associations is related to the significant intensity decrease observed in the very low-frequency range 4 h after milling, reflecting a substantial ordering. Spectra collected 96 h after the milling procedure correspond well to the low- and high-frequency spectra of the anhydrous form, without the signature of water molecules. This study reveals that the co-crystal formation is triggered by very weak H-bonding between CA and TP molecules in the absence of water molecules.

### 2.4. Hydration Stability

Hydration stability experiments were performed on the two types of co-crystals obtained by co-milling. The hydrate form 2 does not undergo any transformation after being exposed to 98% RH for seven days. Raman investigations were performed on the co-crystal form 1′ in situ in the RH cell. Spectra were collected in the high-frequency range, and plotted in Figure 10 at various times of exposure to 90% RH. Figure 10a shows a quasi-instantaneous transformation of the anhydrous form toward the hydrate form, which can be identified by the emerging C–H stretching band at a frequency distinctive to the hydrate co-crystal, at the expense of the band located around 3120 cm^−1^. The time dependence of this spectral region plotted in Figure 10b shows a drastic slowdown of the transformation. This transformation was completed after 72 h of exposure to 90% RH.

## 3. Discussion

The present study demonstrates the very high performance of Raman spectroscopy for analyzing the physical state of organic molecular materials, their structural organization and their subsequent transformations. Raman spectroscopy is very well suited to the analysis of molecular associations via H-bonding, as are other spectroscopies (Infrared, NMR), and the structural organization in the long-range order via low-frequency investigations. Although it is only an indirect probe of the crystalline structure, Raman spectroscopy appears better suited than X-ray diffraction to finely analyze the physical state and the structural organization via H-bonding molecular association and its transformations. As a consequence, using Raman spectroscopy for analyzing theophylline–citric acid co-crystals has provided new findings that are not completely in agreement with previous results obtained from X-ray diffraction experiments [9], since it was shown in the present study that co-milling anhydrous reactants for 60 min did not directly produce co-crystals. Additionally, it was also shown that the neat co-milling of hydrate and anhydrous reactants produces form 1′ which transforms rapidly into the hydrate form 2 when exposed to 90% RH.

In the present study, form 2 can be identified as hydrate 2 in the study of Karki et al. [9], and the assignment of the high-frequency bands involved in H-bonding in the present study can be discussed on the basis of the structural description of hydrate 2 [9]. Some considerations about molecular interactions arising from these structural refinements obtained from single crystal X-ray data [9] should be summarized. There is no H-bond between CA and imidazole nitrogen atom of TP contrarily to other TP and carboxylic co-crystals. Only O–H … O H-bonds between TP and CA are possible. Additionally, water molecules can form O–H … N H-bonds with TP and also interact with CA via O–H … O H-bonds. From these considerations and the analysis of Figure 2c, it can be confirmed that (i) the broad and weakly intense band located around 3300 cm^−1^ corresponds to O–H … O H-bonds between CA and H_2_O/TP, and (ii) the 3360 cm^−1^ corresponds to O–H stretching vibrations involved in H-bonding between TP and H_2_O/CA. It was observed that these bands disappear upon heating during the phase transformation of form 2 into form 1 and are not present in the spectrum of form 1′, suggesting that these bands reflect interactions between water and CA/TP. The Raman band located at lower frequencies (3135 cm^−1^) corresponds to C–H stretching vibrations in TP [22] and is distinctive to form 2. The negative temperature dependence of its position upon heating (Figure 7a) shows the absence of C–H … O H-bonding in form 2 while these H-bonds were observed between TP and water in TPh [22]. This also indicates the absence of C–H … O between TP and CA molecules. The temperature independence of the band frequency after transformation into form 1 around 100 °C, indicates the absence of C–H … O H-bonds between TP and water, confirming the transformation of the hydrate (form 2) into the anhydrous form 1.

It is worth noting that no structural data has been yet reported for the anhydrous co-crystal. This form was obtained either by heating form 1′ or by heating a co-amorphous TP-CA blend. The low-frequency spectra of the three forms are plotted in Figure 11. This spectral region corresponds to the domain of lattice modes reflecting the crystalline fingerprints [30,31,32] and is generally used to identify polymorphic forms. Figure 11 shows that forms 1 and 1′ are almost similar with the exception of the two subtle shoulders existing in the spectrum of form 1′ (localized by arrows in Figure 11), not existing in form 1 and corresponding to the well-defined Raman bands in form 2. These bands can be assigned to molecular interactions between water molecules and TP or/and CA molecules. A quick look at Figure 2a indicates a correspondence between these bands and those detected in CAh at the same frequencies, suggesting that they reflect molecular interactions between water and citric acid within co-crystal hydrate. The low-frequency spectrum is recognized as very sensitive for probing lattice vibrations distinctive to polymorphic forms [32], and can be used in the present study for identifying the pseudo polymorphic form 1′. Except these two shoulders, the low-frequency spectrum of form 1′ is mimicking the spectrum of form 1, indicating similar structural organizations in both forms. The very slight spectral modifications detected upon heating form 1′ confirms that structural descriptions of forms 1′ and 1 are very similar at room temperature. In addition, the molecular interactions responsible for the stability of both forms are also similar and mostly related to the 3510 cm^−1^ band. The temperature dependence of this band plotted in Appendix A indicates that the corresponding OH group is involved in a very weak H-bond between TP and CA, given the high frequency of the band position detected in the domain of free OH vibrations in liquids. The various high-frequency data obtained from the in situ analyses during the various routes of synthesis of form 1 are converging into the consideration of π–H molecular interactions between TP and CA as mostly responsible for the co-crystalline cohesion. The analysis of form 1′ upon heating shows a marginal stability, since first modifications of lattice modes can be detected from about 35 °C (Figure 6b) and the low-frequency shoulders that are distinctive in the presence of water are observed no more at 50 °C (see the arrow in Figure 3a). By contrast, the first lattice modes modifications in form 2 are detected from about 75 °C (Figure 4a and Figure 6b,c). Additionally, an almost instantaneous transformation of form 1′ into form 2 was observed when exposed to 90% RH. This indicates a pronounced metastable character of form 1′ that can easily transform toward form 1 or form 2 if the co-crystal is exposed to weak fluctuations (T or RH) of the room conditions. This metastable character of form 1′ can be associated with the disorder detected via the broadened band-shape of the low-frequency spectrum (Figure 11) compared with form 2, also observed in form 1. This disorder can be considered as inherent to the weak strength of the intermolecular forces. The LFRS of form 1′ can be almost considered as an envelope of the lattice mode spectrum of form 2, suggesting a similar crystalline lattice in the three pseudo polymorphic forms. Powder X-ray diffraction should be performed using synchrotron radiation for obtaining robust data making an accurate structural description of the co-crystal form 1 possible.

Identifying two pseudo polymorphic forms containing different proportions of water molecules is not actually surprising. Indeed, co-crystals obtained by (i) liquid-assisted grinding of anhydrous reactants or (ii) by grinding at least one hydrate reactant with the other should lead to co-crystals containing various proportions of water, different from those corresponding to the stoichiometric ratio of TP, CA and water (1:1:1) within the hydrate form which has been analyzed by Karki et al. [9]. It is also worth mentioning that liquid-assisted grinding could lead to the formation of co-crystals characterized by various degrees of metastability depending on the amount of water used in the preparation with respect to the mass of reactants.

## 4. Material and Methods

### 4.1. Materials

Anhydrous theophylline (C_7_H_8_N_4_O_2_, purity ≥ 99%, form II) and citric acid (C_6_H_8_O_7_, purity ≥ 99.5%) were purchased from Sigma Aldrich (St. Louis, MO, USA), as well as hydrate form (purity ≥ 99%). The hydrate form of theophylline was prepared by exposing the anhydrous form to 98% RH for 24 h.

### 4.2. Methods

#### 4.2.1. Co-Milling

Equimolar concentration mixtures of API (TP) and co-former (CA) typically corresponding to a total mass of about 400 mg were placed in an Eppendorf container for milling using an oscillating milling device (MM400—Retsch) (GmbH & Co., Haan, Germany) for 30 min and one stainless steel ball (Ø = 7 mm), at 30 Hz, for all multicomponent systems (using conditions required for co-crystallization reported by Karki et al. [9]). A procedure alternating milling periods of 5 min with pause periods of 1 min was systematically used for limiting the mechanical heating of the powder.

Cryogenic co-milling was also performed at 30 Hz for 30 min using a Retsch CryoMill (Retsch GmbH & Co., Haan, Germany). Mixtures of typical mass of 1 g were placed in a 50 mL ZrO_2_ jar and milled at −196 °C using one ball (Ø = 20 mm) of the same material. A procedure alternating milling periods of 5 min with pause periods (milling at 5 Hz) of 1 min was also used.

Relative humidity (RH) experiments were performed in situ in the optical cell GenRH-Mcell coupled to the RH generator GenRH-A purchased from Surface Measurement Systems Ltd.( Surface Measurement Systems Ltd, London, UK) In-sit hydration kinetics were performed at 90% RH.

Experiments were also performed by exposing prepared powders to 98% RH in desiccators containing a saturated solution of potassium sulfate.

#### 4.2.2. Raman Spectroscopy

***Low-frequency Raman investigations*** were performed with a very high-dispersive XY Dilor spectrometer (Dilor, Lille, France) composed of three gratings (1800 g/mm). Using the 660 nm line from a Cobolt laser and maintaining the slits opened at 150 µm made it possible to detect a Raman signal down to 5 cm^−1^ in high resolution configuration (lower than 1 cm^−1^). Low-frequency Raman spectra (LFRS) have been collected between 5 and 350 cm^−1^ in 1 min. LFRS have been collected in situ during heating ramp at 1 °C/min. The analysis of the LFRS requires a specific processing inherent to the influence of the temperature on the Raman intensity via the Bose factor [31,33]. This factor induces distortion of the Raman spectrum mostly in the very low-frequency region (*ω* < 50 cm^−1^) which is temperature dependent. Consequently, the low-frequency Raman intensity $I\left(\omega ,T\right)$ is converted into reduced intensity $I_{r}\left(\omega \right)$ via:(1)$$I_{r}\left(\omega \right)=\frac{I_{Raman}\left(\omega ,T\right)}{\left[n\left(\omega ,T\right)+1\right]\omega }$$
where *n*(*ω*,*T*) is the bose factor. This representation of the LFRS is generally used for highlighting molecular disorder corresponding to fast molecular motions thermally activated, detected in the low-frequency region (5–50 cm^−1^) and considered as relaxational motions giving a contribution to the LFRS called quasielastic intensity [31]. In disordered molecular systems, the structural information is contained into the pure vibrational spectrum. It is obtained by removing the contribution of the quasielastic intensity from the *I_r_*(ω)-spectrum which is converted into Raman susceptibility according to:(2)$$\chi \prime \prime \left(\omega \right)=\omega .I_{r}\left(\omega \right)=\frac{C\left(\omega \right)}{\omega }G\left(\omega \right)$$
where *C*(*ω*) is the coupling coefficient between light and vibration and *G*(*ω*) the vibrational density of states (VDOS). *Χ*″(*ω*) is recognized to be a representation very close to the VDOS [34] and is used for obtaining structural information in molecular disordered systems [31]. In the present study, the low-frequency spectra of TP-CA two-component systems were presented in Raman susceptibility, while the temperature dependences of LFRS were presented in reduced intensity to monitor disordering phenomena.

***High-frequency Raman investigations*** were carried out on the InVia Renishaw spectrometer (Renishaw, Gloucester, UK). The 514.5 nm line emitted from a Fandango Cobolt laser was focused on the powder sample via an achromatic lens providing the signal within a large volume of about 1 mm^3^. The sample temperature was controlled by placing the sample in a THMS 600 Linkam temperature device. Spectra were collected in the 2800–3800 cm^−1^ region. The acquisition time of spectra was 1 min, and they were collected in situ during heating ramps at 1 °C/min.

## 5. Conclusions

This study has revealed the existence of three pseudo polymorphic forms. It was shown that the hydrate (form 2) and the anhydrous (form 1) are characterized by two types of H-bonding ensuring the co-crystalline cohesion and stability, depending on the amount of water present in the synthesis method. Co-crystallization in the absence of water is only possible from a disordered state because of the very weak H-bonding molecular interactions between CA and TP. In the presence of a large amount of water, co-crystal hydrate can be formed via strong H-bonds between water molecules and the reactants. Between the two cases, in the presence of a small amount of water, there is co-crystallization in a form resembling the anhydrous form, via the same weak H-bonding between TP and CA, assisted to a lesser extent by H-bonding distinctive of the hydrate.

## Figures and Tables

**Figure 1 molecules-28-01605-f001:**
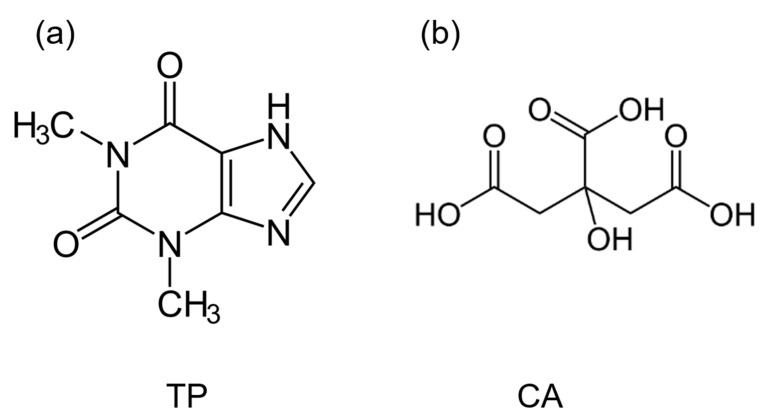
Molecular structures of (**a**) theophylline (TP), (**b**) Citric acid (CA).

**Figure 2 molecules-28-01605-f002:**
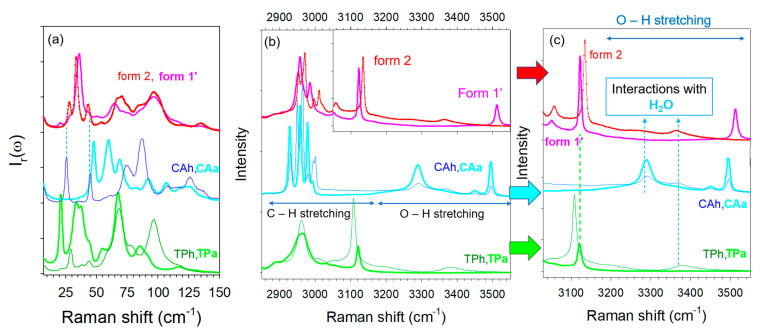
Raman spectra of the two pseudo polymorphic forms of co-crystals, form 1′, form 2, and single components: anhydrous and hydrated theophylline (TPa, TPh), anydrous and hydrated citric acid (CAa, CAh), (**a**) in the low-frequency region, (**b**) in the C–X stretching region (X = C, O), (**c**) in the O–H stretching region.

**Figure 3 molecules-28-01605-f003:**
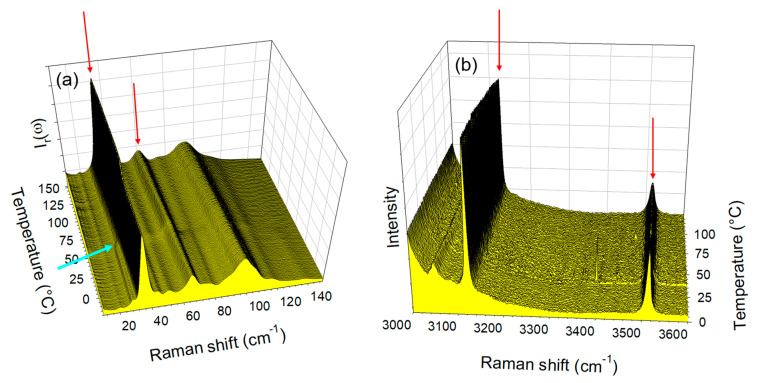
Raman spectra collected in situ upon heating the co-crystal form 1′ at 1°/min (**a**) in the low-frequency region, (**b**) in the O–H stretching region. Raman bands localized by arrows were subsequently analyzed. The blue arrow localizes the disappearance of the 28 cm^−1^ shoulder.

**Figure 4 molecules-28-01605-f004:**
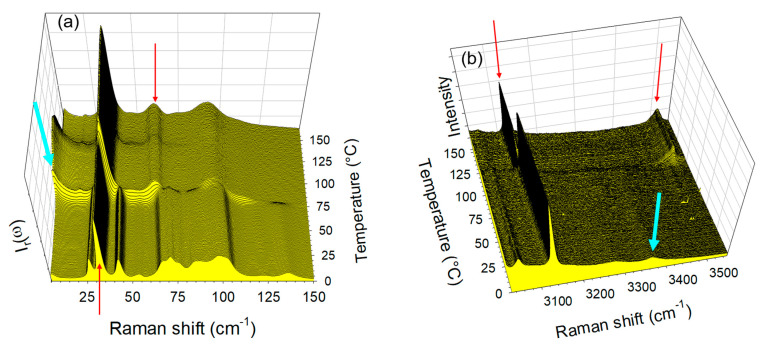
Raman spectra collected in situ upon heating the co-crystal hydrate form 2, at 1 °/min (**a**) in the low-frequency region, (**b**) in the O–H stretching region. Raman bands localized by arrows were subsequently analyzed.

**Figure 5 molecules-28-01605-f005:**
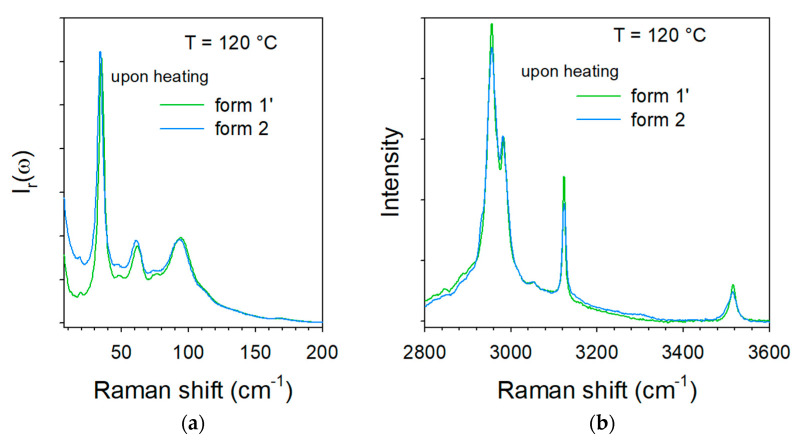
Raman spectra collected at 120 °C upon heating the two pseudo polymorphic forms 1′ and 2 (**a**) in the low-frequency region, (**b**) in the C–H and O–H stretching region.

**Figure 6 molecules-28-01605-f006:**
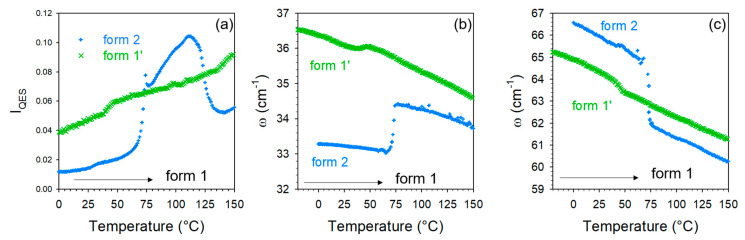
Data analysis of the low-frequency spectra of the co-crystal form 1′ and 2; (**a**) temperature dependence of the quasielastic intensity (I_QES_); (**b**) temperature dependence of the position of the phonon peak located around 35 cm^−1^; (**c**) temperature dependence of the position of the Raman band located around 66 cm^−1^.

**Figure 7 molecules-28-01605-f007:**
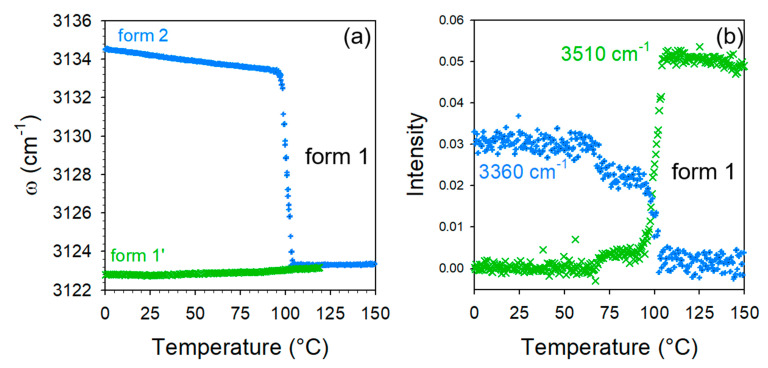
Data analysis of the high-frequency spectra of the co-crystal forms 1′ and 2; (**a**) temperature dependence of the C–H stretching band distinctive of forms 1′ and 2; (**b**) Integrated intensity of the O–H stretching bands distinctive of molecular associations in the co-crystal form 1′ (3510 cm^−1^) and in the co-crystal hydrate form 2 (3360 cm^−1^), measured upon heating form 2.

**Figure 8 molecules-28-01605-f008:**
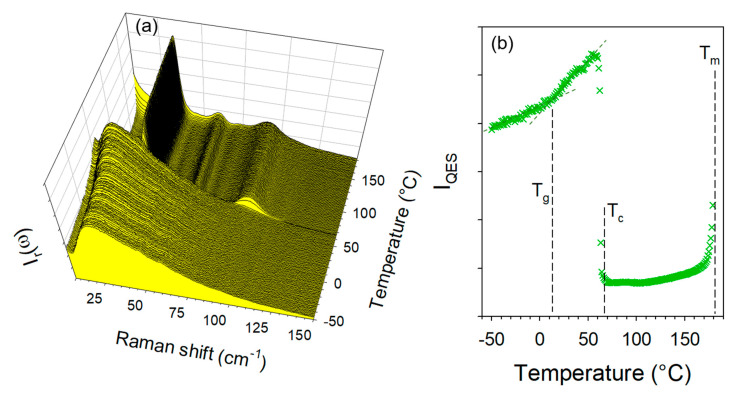
Low-frequency analysis of the heating run (1 °C/min) of the co-amorphous formulation obtained by co-milling anhydrous forms of theophylline and citric acid; (**a**) temperature dependence of the low-frequency spectrum; (**b**) temperature dependence of the quasielastic intensity obtained by integrating I_r_(ω) spectra in the very low-frequency region (<30 cm^−1^).

**Figure 9 molecules-28-01605-f009:**
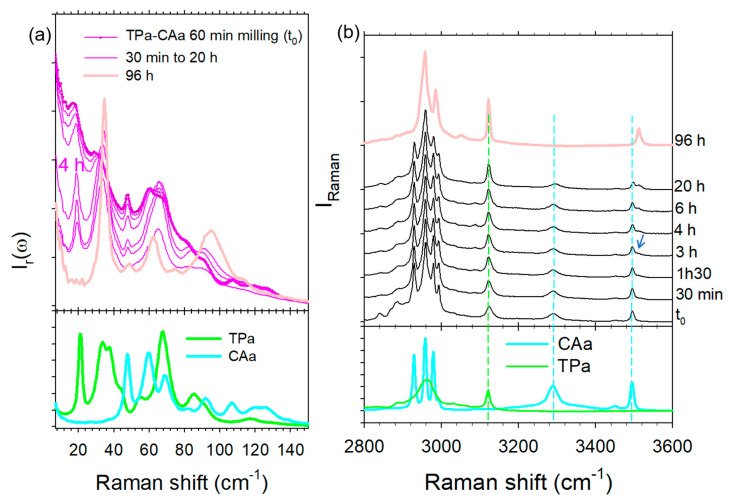
Spectra of (TPa, CAa) mixture co-milled at room temperature for 1 h, collected at various times after the milling procedure; (**a**) in the low-frequency region; (**b**) in the high-frequency region; spectra are compared with those of the anhydrous reactants (bottom graph); spectra were taken and plotted at same times in the low- and high-frequency regions.

**Figure 10 molecules-28-01605-f010:**
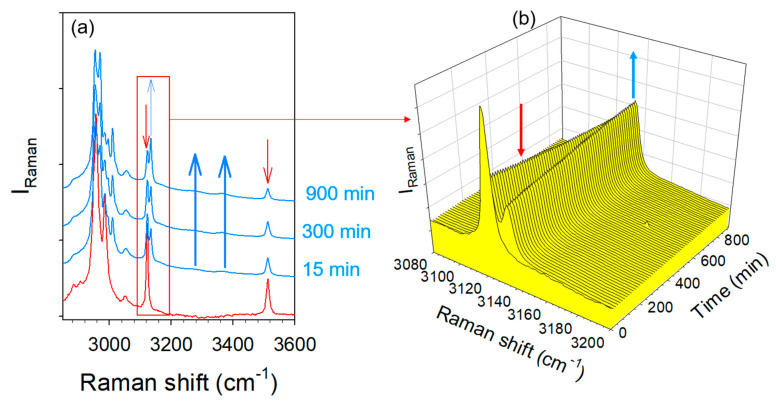
Time dependence of the high-frequency spectrum taken in the co-crystal form 1′ exposed to 90% RH; (**a**) in the C–H and O–H stretching regions, the two blue thick arrows localizing the appearance of the broad O–H stretching bands distinctive of H-bonding with water molecules; (**b**) in the C–H stretching region distinctive of co-crystals forms 1 and 2.

**Figure 11 molecules-28-01605-f011:**
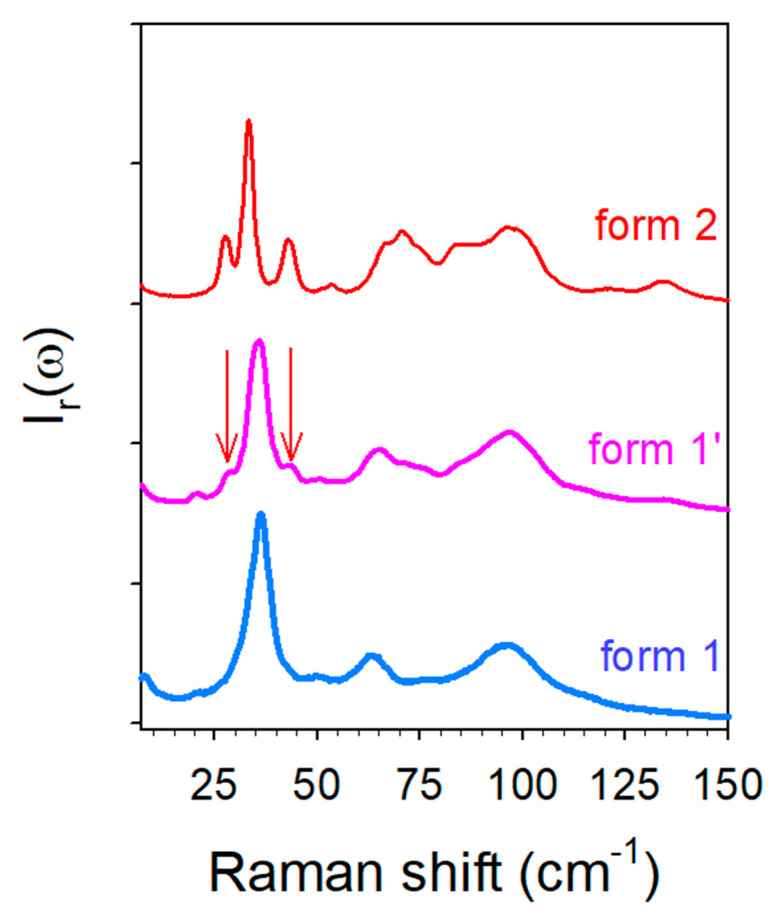
Low-frequency spectra of pseudo polymorphic forms of TP-CA co-crystals at room temperature.

## Data Availability

The data related to this study are presented in the manuscript and Appendix A.

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
