# Peer review of "Analysis of Co-Crystallization Mechanism of Theophylline and Citric Acid from Raman Investigations in Pseudo Polymorphic Forms Obtained by Different Synthesis Methods"

_molecules, 2023, doi:10.3390/molecules28041605_

Round 1
Reviewer 1 Report
The objectives of this manuscript are well presented, however, there are a few notable mistakes that need to be corrected. The title of the manuscript needs to be modified. It cannot start with the word “Using”. The language also needs to be revised as there are a number of careless spelling mistakes as well as inconsistencies e.g. on page 1 the word cocrystal and co-crystal appear in the same paragraph. Other words with the same mistakes include co-milling and comilling, cocrystallization and co-crystallization, etc. The Colour coding in Figure 2 is not clear enough to distinguish between the two graphs. From the experimental results in section 2.1 it is somewhat difficult to determine the differences between the hydrated form and anhydrous form without knowing the number of crystal waters attached to CA and TP. Is it feasible to use TGA to determine the number of crystal waters? Generally, a secondary technique other than Raman spectroscopy could have been essential for the verification purposes of the obtained results. Please check your references as some might have typo errors e.g. reference number 5, year of publication 2055? There is a lot of self-citation in this manuscript where authors appear more than 3 times in the used references.
Reviewer 2 Report
Article “Using various routes of synthesis for understanding the co-crystallization mechanism of theophylline and citric acid” by Yannick Guinet, Laurent Paccou and Alain Hédoux presents a study of the formation of theophylline and citric acid co-crystal phases and phase transformation among these phases using Raman spectroscopy methods. The study contains aspects of notable scientific interest, for example, study of the mechanism of the co-crystal formation from co-grinded amorphous phase, using spectroscopic method allowing identification of intermolecular interactions involved in this transition. However, the manuscript contains aspects which makes it unacceptable for publication in the current form.
1. Most importantly, it is the speculation of the intermolecular interactions present in theophylline and citric acid co-crystal and particularly its hydrate. Crystal structure of this hydrate is reported and available in the CCDC (refcode KIGKAN), therefore it is clear what intermolecular interactions are present in this structure, but this information is not used in this study. Instead, the observed peaks are assigned to vibrations and effects of interactions based on general knowledge and some speculations. Among such speculations the presence of π-H bonding in co-crystal hydrate is mentioned multiple times and has a central role in the discussion of the formation of this phase. However, to my understanding, water in theophylline and citric acid co-crystal hydrate forms only conventional hydrogen bonds. Therefore structure analysis and clear crystallographic demonstration of the relevant intermolecular interactions should be included in the manuscript. Additionally, crystals structures are known for numerous theophylline anhydrous co-crystals as well ad citric acid co-crystals, therefore the interpretation of the peaks of anhydrous theophylline and citric acid co-crystal should be complemented by information about the potential interactions based on the already know structures formed by these constituents.
2. Additionally to this, even the presented conclusions about the co-crystallization mechanism are still only speculative without convincing proofs or solid explanations provided for the proposed changes of the interactions during the crystal phase formation or phase transformations. The most obvious among such speculations is weak π-H-bond interactions in anhydrous TP-CA form.
3. Then also authors identify different crystal forms of a) TP, b) CA and c) TP-CA co-crystal from the Raman spectra, but they does not provide information what they are using as a standards form this, as, for example, there are several polymorphs of TP, and in some cased distinction between hydrate and anhydrate form seems ambiguous from the recorded spectra. Moreover, crystal structure for almost all the studied crystal forms are available, which could be used for simulation of the Raman spectra using computational calculations, or other methods allowing clear identification (such as X-ray diffraction) can be used.
Apart from this, there are also numerous additional points which should be taken into account by revising the manuscript:
1. Abstract should summarize the most important findings of the article. Instead, it mostly summarizes what has been done. Also, the authors state in the abstract “This study shows the high capabilities of Raman spectroscopy to discriminate pseudo polymorphic forms and for monitoring their transformations” although this is not the main finding of the article and actually has been clearly shown only for one system (TP-CA).
2. In the Introduction authors draw a picture that the polymorphism in co-crystals is rather unexplored, although there are many studies dealing with this issue, which in fact is rather common and commonly reported. For example, the authors miss already nearly 10 years old review Aitipamula, Chowa and Tan (10.1039/C3CE42008F) clearly presenting numerous examples. Also control of the co-crystal polymorphism using different preparation approaches have been often observed, yet authors cite only a study of caffeine – glutaric acid from 2005 (!). Interestingly, authors also mention example of different forms obtained for caffeine – citric acid cocrystal again citing not the most recent studies, although a recent summary from William Jones group clearly reveals many aspect of this system, including disappearing polymorphism (10.1021/acs.cgd.9b01431).
3. The authors write “It is also well known that TP and CA easily form co-crystals anhydrate or hydrate” and “Given the sensitivity of Raman spectroscopy to detect polymorphic modifications and for detecting H-bonding” but does not provide reference for these statements.
3. The fact that anhydrous TP-CA contain some water should be written otherwise, as the sample cannot be called anhydrous TP-CA. Moreover, from the discussion it is not clear in what form the water is present. Can authors be sure that these are not impurities of hydrate? What are the proof that the “water molecules not or weakly bound to TP or/and CA molecules” and what physical form observable in the Raman spectra and inducing peak shift of the whole structure that would correspond? Considering that the anhydrate sample is either not pure or the form of water is not known and explained, I believe the data and their analysis on heating of anhydrous TP-CA should be transferred to Supporting Information.
4. Moreover, heating of TP-CA is presented in 5 figures (Figure 3 to Figure 7). There is no reason to present all of them in the main text, and most can be transferred to Supporting Information, by keeping clear picture showing changes in Raman spectra by heating hydrate and if unambiguous analysis of peaks are provided also picture demonstrating changes in particular bands by the temperature. Besides, the text does not provide the aim for such heating experiments, as the study was aimed to analyze crystal form obtained using different co-crystal preparation techniques.
5. On P6L14 authors state that there is temperature difference in dehydration from LF and HF spectra. As based on the experimental section the spectra are recorded in separate experiments, the authors should discuss whether this could be because of different sample or sample preparation procedures used.
6. Can the authors discuss what are the proofs that “phase transition mechanism between hydrate and anhydrous forms” are “triggered by vey weak changes in the molecular organization” and that this occur “without change in the crystalline symmetry”?
7. Can the authors explain why the peaks of anhydrous TP are broader (P3L114).
8. Also the aim for performing the cryogenic milling as well as exploration of hydration stability are not clear. Perhaps in the beginning of Results and Discussion section authors should provide what will be done in this study and why these experiments and analysis are performed.
9. The identification of the glass transition temperature in Figure 8 is very ambiguous. The location of this temperature should be clearly demonstrated in the Figure. Moreover, considering the experimental noise in Figure 8b, I believe that only repeated experiments can show that there is in fact glass transition observed during the heating and at what temperature.
10. I once again note that the crystallization from the sample obtained in the co-milling are the most interesting results of this study. However, unambiguous results would require performing additional experiments (as crystallization in general in kinetic phenomena and therefore high sample-to-sample variation is expected). Moreover, as also stated above, interpretation of the results should be based on the available structure information and must include more convincing proofs.
11. The Discussion section should be rewritten. Firstly, almost all first paragraph is information which should be included in Introduction, if relevant, but is definitely not a discussion. Secondly, authors note that “new findings which are not completely in agreement with previous results” but do not explain the differences and discuss the possible reasons. Thirdly, it is not explained what is meant by “hydrate and an anhydrous forms can be obtained with a similar crystalline lattice”. To my understanding, X-ray diffraction experiments are required to obtain such information. Fourthly, 2nd and 3rd paragraphs of this section are actually summary and again no discussion is provided, just summary of the above presented results.
12. Section Conclusions seems to contain sentences more appropriate to Abstract. Therefore – Conclusions could be formed from the present section Discussion and Abstract could be improved by using the current Conclusions as part of it.
13. The English language should also be improved, as there are several typo and many incorrect grammatical structures. However, as there is large amount of such structures and many scientific objections, they are not listed in this review.
Some minor points:
14. Abbreviation LRFS is used without providing full term before the first use.
15. In the experimental section the volume of the grinding jar is not given.
16. Reference 5 has a typo in publication year.
Reviewer 3 Report
This is a neat and thorough study of cocrystals of theophylline and citric acids by low and high frequency Raman spectroscopy. It is a nice paper and worthy of publication.
The only - and not minor - problem with this investigation is the lack of control (inherent to the technique employed) on the structure of the materials. Since the results are discussed also in the light of the results of the structural study cited in ref. 9, one would expect that - at least - powder Xray diffraction patterns were registered and compared. It is not unusual, actually it is quite common, that different preparation procedures lead to different crystal forms (eg hydrates, polymorphs etc.).
I do believe that the authors should - at least - find a way to measure the XRPD of their compounds and compare with those in ref. 9 (whether measured or calculated on the basis of the single crystal structures)
Round 2
Reviewer 2 Report
I still don’t see reasons not to stick with part of the points made in the initial review which authors decided to leave unchanged in the revised version of the manuscript.
There is no overlay of Form 1 and the proposed Form 1’ in both studied spectral regions. The authors have changed the term not pure anhydrous TP-CA to Form 1’ and based on the provided information it is hard to evaluate whether this is better or, in contrary – worse, description. Considering the very fast transformation of anhydrous form to hydrate (notable conversion in 15 min at 90% RH), hydrate impurity could easily form. The spectra, however, cannot clearly support the existence of two anhydrous forms, particularly considering that it is not clear what is the role of water in form 1’ and why it has clear effect in low frequency range apparently not in high frequency range. The authors avoid using any other experimental technique, although DSC would be required to characterize and prove the existence of two polymorphs, which could be supplemented by XRPD for additional proof and understanding on similarity or differences in long range structure. TG could provide understanding to the role of water in form 1’.
Then form 1’ is evaluated as metastable, as it transforms into form 1 by heating and form 2 in high humidity. Here stability studies at ambient conditions would be required, as both these conditions clearly will facilitate stability of anhydrous form and hydrate.
The discussion about intermolecular interactions in anhydrous form leaves little space for other potential scenarios, although without crystal structure model in hand the presented results does not convince me that these interactions are the only ones which could result in such spectral lines. Considering the molecular structures of CA and TP in the anhydrous form there should be hydrogen bonds COOH…O=C and most likely also O…NH, COOH…N as in any other structure of co-crystals formed by these or similar compounds. Considering the number of strong hydrogen bonds acceptors and donors present these should be the most characteristic and energetically significant intermolecular interactions. Therefore without crystal structure information I don’t see any physical evidence for highlighting interaction π-H as the most important hydrogen bond.
The article is too long and contains some duplication of results in Figures. The 2D spectra have almost no additional value compared to the plots where the characteristic peaks are extracted in spectral form (as in Figure 10) or in intensity – temperature plots. Although authors claim to demonstrate transformation mechanism using these spectral representations, they only make it harder to clearly see the effect of temperature on peak shapes, positions, and intensities.
I also don’t support the authors’ use of term “co-crystallization mechanism” for basically just finding out under which conditions phase transition occur or what form is obtained in given conditions.
Reviewer 3 Report
molecules-2110698 is now acceptable for publication.
Author Response
No additional revision requested